# Genetic Improvement and Application Practices of Synthetic Hexaploid Wheat

**DOI:** 10.3390/genes14020283

**Published:** 2023-01-21

**Authors:** Hongshen Wan, Fan Yang, Jun Li, Qin Wang, Zehou Liu, Yonglu Tang, Wuyun Yang

**Affiliations:** 1Crop Research Institute, Sichuan Academy of Agricultural Sciences, Chengdu 610066, China; 2Key Laboratory of Wheat Biology and Genetic Improvement on Southwestern China, Ministry of Agriculture and Rural Affairs, Chengdu 610066, China; 3Environment-Friendly Crop Germplasm Innovation and Genetic Improvement Key Laboratory of Sichuan Province, Chengdu 610066, China; 4Biotechnology and Nuclear Technology Research Institute, Sichuan Academy of Agricultural Sciences, Chengdu 610066, China

**Keywords:** synthetic hexaploid wheat, genetic improvement, application practices

## Abstract

Synthetic hexaploid wheat (SHW) is a useful genetic resource that can be used to improve the performance of common wheat by transferring favorable genes from a wide range of tetraploid or diploid donors. From the perspectives of physiology, cultivation, and molecular genetics, the use of SHW has the potential to increase wheat yield. Moreover, genomic variation and recombination were enhanced in newly formed SHW, which could generate more genovariation or new gene combinations compared to ancestral genomes. Accordingly, we presented a breeding strategy for the application of SHW—the ‘large population with limited backcrossing method’—and we pyramided stripe rust resistance and big-spike-related QTLs/genes from SHW into new high-yield cultivars, which represents an important genetic basis of big-spike wheat in southwestern China. For further breeding applications of SHW-derived cultivars, we used the ‘recombinant inbred line-based breeding method’ that combines both phenotypic and genotypic evaluations to pyramid multi-spike and pre-harvest sprouting resistance QTLs/genes from other germplasms to SHW-derived cultivars; consequently, we created record-breaking high-yield wheat in southwestern China. To meet upcoming environmental challenges and continuous global demand for wheat production, SHW with broad genetic resources from wild donor species will play a major role in wheat breeding.

## 1. Introduction

Even though wheat (*Triticum aestivum* L.) is the most widely consumed food in the world, its global production still needs to be enhanced to meet the growing demand in the coming decades [1]. To improve the genetic yield potential of wheat, the introduction of alien genomic components with superior alleles and elite genes from wild/related species is considered an effective conventional method that could broaden the narrow genetic basis of modern common wheat, especially the D genome [2]. However, homologous recombination between alien chromatin and wheat chromosomes is often repressed [3,4,5]; alien chromatin also leads to chromosomal aberrations [6], which is disadvantageous for breeding.

To simulate and study the origin process of the common wheat, synthetic hexaploid wheat (SHW) was created by crossing tetraploid *T. turgidum* with *Aegilops tauschii* and subsequently doubling chromosomes [7,8,9,10]. Fortunately, SHW makes it easy to transfer both *T. turgidum* and *Ae. tauschii* genes into common wheat, and homologous recombination could break the undesirable gene linkages when crossing SHW with common wheat and fix desirable haplotypes. Therefore, the number of released elite commercial wheat cultivars derived from SHW has continually increased [11,12,13,14,15].

## 2. High Breeding Potential of Synthetic Hexaploid Wheat

It is thought that the use of SHW can increase wheat yield from the perspectives of physiology and cultivation [16,17,18] because SHW is thought to be a source of genetic diversity for important physiological traits such as enhanced photosynthetic rate [19]. Among the most representative and widely planted cultivars released from 1969 to 2012 in southwestern China, the SHW-derived cultivars possessed higher values of physiological traits, including dry matter accumulation at maturity, harvest index, N utilization efficiency, soil and plant analyzer development (SPAD) value, and canopy/net photosynthetic rate, compared to non-SHW-derived cultivars [18]. This was especially true for SPAD value, canopy apparent photosynthesis rates, and harvest index [16,17], which showed an 11.5% yield increase [16]. For phosphorus use efficiency, SHW-derived cultivars were rated as efficient compared to non-SHW-derived cultivars, which were determined to be moderately efficient or inefficient according to the phosphorus efficiency index inferred from principal component analysis and cluster analysis [20]. Moreover, the use of SHW has the potential to improve a range of stress-adaptive traits of modern bread wheat, such as increased water use efficiency under drought conditions [21].

From the perspective of genetic improvement, favorable genes resistant to biotic/abiotic stresses and related to yield and yield components in *T. turgidum* or *Ae. tauschii* are normally expressed in a hexaploid genetic background, and many reported QTLs/genes have been mapped to the AB and D genomes of SHW. The genes that were reported to be resistant to rust [22,23,24], powdery mildew (*Pm34* [25]; *Pm58* [26,27]), pests (*H26* resistant to Hessian fly [28]; greenbug resistance genes [29]), aluminum [30], and drought [31] from *Ae. tauschii* were also transferred to common wheat, and serve as a gene reservoir for modern wheat adaptation [32,33].

For yield-related traits, using the common wheat ‘Flair’ [34], ‘Opata85’ [35], and ‘Chuanmai 32’ [36] as controls, favorable QTL alleles were also detected in the AB and D genomes of SHW lines ‘XX86’, ‘W7984’, and ‘SHW-L1’; the alleles are mostly involved in grain weight, grain number per spike, spike length, and tiller number. Wan et al. [37] detected a major QTL for leaf sheath hairiness (LSH) that was also associated with grain yield in the common wheat chromosome 4DL; the favorable QTL allele was introgressed from *Ae. tauschii*, which enhanced grain yield by increasing grain weight. This finding indicated that SHWs might carry yield-related QTL alleles superior to those of modern common wheat.

## 3. Enhanced Genomic Variation and Recombination in Synthetic Hexaploid Wheat

Approximately 9000 years ago, an accidental hybridization between domesticated emmer wheat (*T. turgidum* conv. *turgidum*, 2n = 4*x* = 28) and goat grass (*Ae. tauschii* spp. *strangulate*, 2n = 2*x* = 14) with chromosome doubling naturally resulted in the generation of a free-threshing hexaploid common wheat (*T. aestivum*, 2n = 6*x* = 42) (Figure 1). This hybrid accounts for approximately 95% of current global wheat production, with tetraploid *durum* wheat (*T. turgidum* ssp. *durum*) representing the remaining 5% [38]. Allohexaploidization added the *Ae. tauschii* D genome into tetraploid wheat, and the allohexaploid wheat was more adaptive to changing environments and then spreading more rapidly around the world than the tetraploid wheat.

However, the D genome of the first bread wheat originated from only a small number of wild *Ae. tauschii* ssp. *strangulata* plants [2,39], and the AB genome was suggested to be from a free-threshing form of tetraploid wheat [39,40]. As the AB and D genomes originated from limited sources, the individuals from tetraploid and diploid parents involved in the hexaploidization of wheat did not possess all superior characteristics in a few totipotent plants to make them sufficiently adaptable to changing environments worldwide. Therefore, in addition to the advantages of heterosis and gene redundancy [41], there are likely other forces in hexaploidization that accelerate wheat evolution and spread.

Our lab simulated the evolutionary hexaploidization process and generated SHW using different *Ae. tauschii* and tetraploid wheat with the ability to automatically double chromosomes, which occurs by unreduced gamete formation controlled by genetic factors (*QTug.sau-3B* [42]). With the 10× resequencing data of SHWs and their parents from next-generation sequencing (NGS) technology, we found that the sequences of the tetraploid AB and diploid D genomes were altered in SHW (Table 1: unpublished data provided by H.W.), and kept changing in subsequent generations of SHW.

Genomic change by DNA elimination and interchromosomal exchange often occurs in newly formed hexaploid wheat [43,44,45]. Wan et al. [9] found that approximately 10% of the SNP loci of *Ae. tauschii* were eliminated in derived SHW using diversity array technology (DArT)-Seq technology. Moreover, in the co-dominant genotypes of F_2_ individuals from a diploid population (SQ665 × SQ783, 2*x*, D_1_D_1_ × D_2_D_2_) and a new SHW-derived population (Langdon/SQ665 × Langdon/SQ783, 6*x*, AABBD_1_D_1_ × AABBD_2_D_2_), the recombination frequency of *Ae. tauschii* was found to be enhanced 2.3-fold by hexaploidization with *T. turgidum* [9].

The changes of ancestral genomes during hexaploidization could generate more genovariation or new alleles, and the gene redundancy that contributed to polyploidization could shield polyploids from the deleterious effect of unfavorable genomic variations. Additionally, the increased genetic recombination in new SHW-derived cultivars could produce more new allelic combinations subject to natural or artificial selection; the evolution of wheat could be accelerated via hexaploidization, and this could help wheat to rapidly spread and increase its role as a major global crop. Therefore, SHW have the potential to enhance variation and adaptive evolution of bread wheat in the breeding process [13].

## 4. A Case of Successful Direct Application of SHW: Chuanmai 42 from Southwestern China

SHW enhanced genomic variation and recombination, and had high breeding potential. However, primary SHWs also have many unfavorable traits, such as late maturity, taller plants, and difficulty in threshing, which made direct application of SHW difficult. Therefore, our team presented a breeding strategy using a large population with limited backcrossing to common wheat (Figure 2). This breeding strategy involved three core aspects: (1) limited backcrossing with common wheat (2–3 times), reserving more favorable genes and genetic diversity from SHW; (2) using a population of more than 1000 individuals to select favorable gene recombination events between SHW and the backcrossed common wheat; and (3) selecting agronomic traits under multiple environments and testing the candidates’ yield potential using high-yield cultivation methods.

Under this breeding strategy, our team used one SHW line introduced from the International Maize and Wheat Improvement Center (CIMMYT), Syn769, successively crossed with two local common wheat lines, SW3243 and Chuan 6415, and three cultivars were bred: Chuanmai 38, Chuanmai 42, and Chuanmai 43. Among them, Chuanmai 42 was released in 2003 and was the first commercial SHW derivative in the world. In the Sichuan regional trials of wheat cultivars in 2002 and 2003, the average grain yield of Chuanmai 42 increased by 70.2% and 28.3%, respectively, compared to the check cultivars Chuanmai 28 and Chuanmai 107 (Table 2). Overall, grain yield increased by 35%, which broke the yield record of commercial cultivars in southwestern China [46,47].

Since 2003, the use of Chuanmai 42 in wheat production has increased grain output by approximately 1,000,000,000 Kg compared to the old cultivars [47]. Moreover, as a leading wheat cultivar, Chuanmai 42 has become a foundation breeding parent for wheat improvement in southwestern China. From 2008 to 2021, a total of 26 cultivars were selected from crosses containing Chuanmai 42; among them, 17 cultivars were from the first generation of Chuanmai 42 crossed with another parental line with high-yield potential, such as Chuanmai 104 and Chuanmai 602 (Table 3). The application of the SHW-derived Chuanmai 42 received a Second-class Prize of the State Scientific and Technological Progress Award in 2010 and a First-class Prize of the Sichuan Province Scientific and Technological Progress Award in 2009.

Since 2003, why is Chuanmai 42 so widely applied in wheat production and breeding in southwestern China? Additionally, what is the role of the SHW germplasm in Chuanmai 42 production?

Unlike the common hexaploid wheat with glabrous leaf sheaths, most SHW accessions have a hairy leaf sheath, which is mostly present in wild species of *Triticeae*, such as *Ae. tauschii* and *T. turgidum* var. *dicoccoides*. Chuanmai 42 has a hairy leaf sheath that was derived from the SHW line Syn769 (Figure 3). Genetic analysis showed that the LSH is from *Ae. tauschii* with its controlling gene on chromosome 4DL [37]. Interestingly, almost all cultivars derived from Chuanmai 42 inherited the hairy leaf sheath character, even in the 2nd generation, which was caused by the tight linkage between the LSH gene and the QTLs associated with grain weight and yield in Syn769 (Figure 3E; [37]). In breeding programs, the leaf sheath hairiness can be used as a morphological marker for high grain yield QTL selection from Chuanmai 42 derivatives.

With a total of 1029 simple sequence repeat (SSR) and 2268 DArT markers detecting polymorphisms among three parents (Syn769, SW3243, and Chuan6415) of Chuanmai 42, the frequency of SHW alleles introgressed to Chuanmai 42 was 15.14%, which was significantly less than the expected 25% assuming random gene assortment. The distribution of introgressed alleles over the A, B, and D genomes was not uniform (B > A > D); introgression occurred most frequently on chromosomes 1A, 1B, 2B, 3A, 4D, 6A, and 6B, whereas none were detected on chromosomes 1D and 7A [13,48,49]. On chromosome 1B, the stripe rust resistance gene *YrCH42* from Chuanmai 42 (Figure 4A; [50]) and the QTL allele for increased grain number per spike (Figure 4B; [51]) originated from its SHW parent Syn769 (Figure 4C; [48]).

## 5. Further Application Using SHW-Derived Cultivars: Chuanmai 104

In 2020, the grain yield of Chuanmai 104 reached 729.8 Kg/mu in the high-yield cultivation field in Jiangyou, Sichuan, which was a record for the highest yield of wheat in southwestern China [52,53]. It was selected from the 127 F_7_ recombinant inbred lines (RILs) of an SHW-derived Chuanmai 42 crossed with Chuannong 16 (Figure 5), which increased the grain yield by approximately 8.42% more than Chuanmai 42 in 2010–2012 national regional cultivar trials.

We genotyped each F_7_ RIL using SSR and DArT markers, and evaluated yield-related traits of each RIL (10 m^2^/plot) in multiple environments with high-yield cultivation. The F_2_ population of Chuanmai 42 × Chuannong 16 was exposed to different environmental stress to generate more gene recombination [54,55], and 127 RILs were finally obtained. Additionally, we combined phenotypic and genotypic evaluations to artificially select cultivars with high yield potential. Our team referred to this as the ‘RIL-based breeding method’ (Figure 5). Their obvious differences with the traditional phenotypic evaluation for QTL analysis are reflected in the following aspects: (1) replacing single-seed sowing with planting density of high-yield wheat production in China, considering the trade-off between yield components and interplant competition; (2) expanding the planting area of each plot from 2–3 rows to at least 10 m^2^, simulating field conditions of wheat production, with the aim to bridge the gaps between individual and population performances. In addition, evaluating the contributions of high-yielding QTLs to grain yield on the population level rather than in the individual plant or panicle level was also highly recommended by Xiong et al. [56] recently.

The yield potential of Chuanmai 104 was enhanced by its pre-harvest sprouting resistance and low-temperature tolerance at the flowering stage (Figure 6: unpublished data provided by Y.T.); it is more resistant than its parent, Chuanmai 42 (Figure 6B; [57]). Moreover, Chuanmai 104 had stripe rust and powdery mildew resistance, inheriting the resistance loci of *YrCH42* [50], *Qyr.saas-7B* [58], and *QPm.saas-4AS* [59] from its parent Chuanmai 42 [51]. The actual grain yield over five continuous years’ field production ranged from 650 kg/mu to 700 kg/mu.

Chuanmai 104 was listed as the leading commercial cultivar for wheat production by the Sichuan Province government from 2013 to 2016 and the Ministry of Agriculture and Rural Affairs of China from 2015 to 2016. From 2015 to present, a total of 11 cultivars were selected from crosses involving Chuanmai 104, such as the big-spike cultivar Chuanmai 93 and tight-plant-type cultivar Chuanmai 98; generally, Chuanmai 104 has also been used as a foundation parent for recent wheat breeding (Table 4). Since 2012, the planting acreage of Chuanmai 104 has totaled 21.5 million mu and generated an additional economic output of 2.4 billion Chinese yuan [53]. The breeding and application of Chuanmai 104 obtained the First-class Prize of the Sichuan Province Scientific and Technological Progress Award in 2020.

The genetic components of Chuanmai 104 were dissected by QTL mapping for yield-related traits [51,60,61], pre-harvest sprouting resistance [57], and powdery mildew resistance [59]. On chromosome 1B, the majority of genomic regions of Chuanmai 104 were from the female parent Chuanmai 42, and among the detected QTLs or genes, the QTL allele of Chuanmai 42 increased the resistance of stripe rust and grain weight (Figure 7). On chromosome 1DS, the genomic region of Chuanmai 104 associated with increasing spike number per m^2^ and grain yield was from Chuannong 16. Chromosome 2BS of Chuanmai 104 was from Chuannong 16, and is related to enhanced grain yield and pre-harvest sprouting resistance, whereas its chromosome 2BL was from Chuanmai 42 and was associated with increased thousand-kernel weight and plant height. The chromosome arm 4AS of Chuanmai 104 was from Chuanmai 42, and QTL alleles from Chuanmai 42 increased the grain number per spike and powdery mildew resistance (Figure 7; [59]). For 4DS, the QTL alleles of Chuanmai 42 increased thousand-kernel weight, grain number per spike, and spike number per plant, finally enhancing grain yield (Figure 7; [37,49,51,60]), and Chuanmai 104 inherited the favorable QTL alleles from Chuanmai 42. For the yield-related QTLs on chromosome 5B and 5D, Chuanmai 104 inherited the QTL alleles from Chuanmai 42 with increased thousand-kernel weight and grain yield, respectively (Figure 7; [51]).

Chuanmai 104 inherited the big-spike properties of higher grain number per spike and grain weight from SHW-derived Chuanmai 42, multiple spike properties including spike number per square meter from Chuannong 16, and grain yield from both parents contributed. Additionally, Chuanmai 42 provided stripe rust [50,58] and powdery mildew resistance [59], whereas Chuannong 16 provided pre-harvest sprouting resistance [57].

## 6. Conclusions and Perspectives

SHW is a useful genetic resource that can be used to improve the performance of common wheat by transferring favorable genes from a wide range of tetraploid or diploid donors because many QTLs/genes can be expressed under a genetic background of common hexaploid breeding lines [34,35,36,62,63,64,65]. The use of SHW could also bypass the disadvantages of using alien chromatin such as from rye, *Agropyron elongatum*, and *Haynaldia villosa*, which undergo reduced homologous recombination with wheat chromosomes. Importantly, genomic variation and recombination were enhanced in newly formed SHW, which could generate more genovariation or new gene combinations compared to ancestral genomes. Additionally, the genomic instability of newly formed SHW can also enhance the genetic variation and recombination when they are crossed with common breeding lines, as well as abundant variation in agronomic traits [66].

How to utilize primary SHW for wheat improvement effectively? Different geneticists and/or breeders summarized various advisable strategies from different perspectives [15,67]. Here, according to our application practices of SHW for decades, using the ‘large population with limited backcrossing method’ was suggested as an effective breeding strategy for direct breeding application of SHW to improve the breeding population and select elite cultivars. These improved SHW derivatives could be used as a basic genetic framework for the next round of pyramiding favorable genes from new SHW lines or other breeding lines. Additionally, for further QTL/gene pyramiding, the RIL-based breeding method could be beneficial by accurately evaluating their breeding value and identifying suitable new cultivars that should be selected.

The global demand for wheat production will keep continuously growing with the world’s increasing population, especially as global warming becomes an increasing worldwide threat that enhances difficulty in crop cultivation. In this situation, a significant increase in yield like during the Green Revolution is desirable to break up the bottleneck of wheat yield and make wheat cultivars tolerant to drought, heat, cold, and flooding stress. SHW along with broad genetic resources from wild donor species will play a big role in the race to meet upcoming environmental challenges and the continuous global demand for wheat production.

## Figures and Tables

**Figure 1 genes-14-00283-f001:**
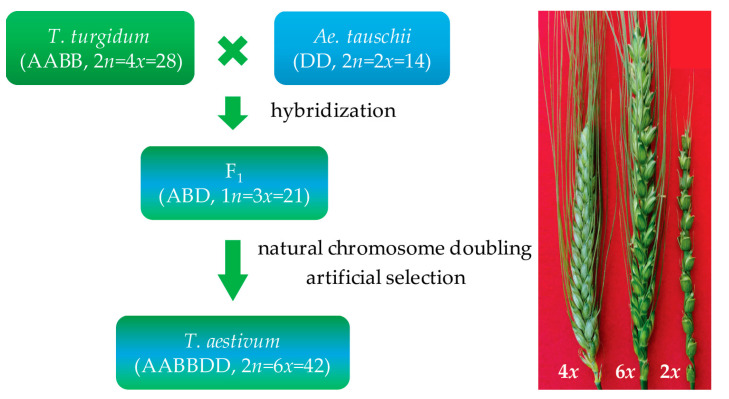
Hexaploidization and artificial synthesis of wheat.

**Figure 2 genes-14-00283-f002:**
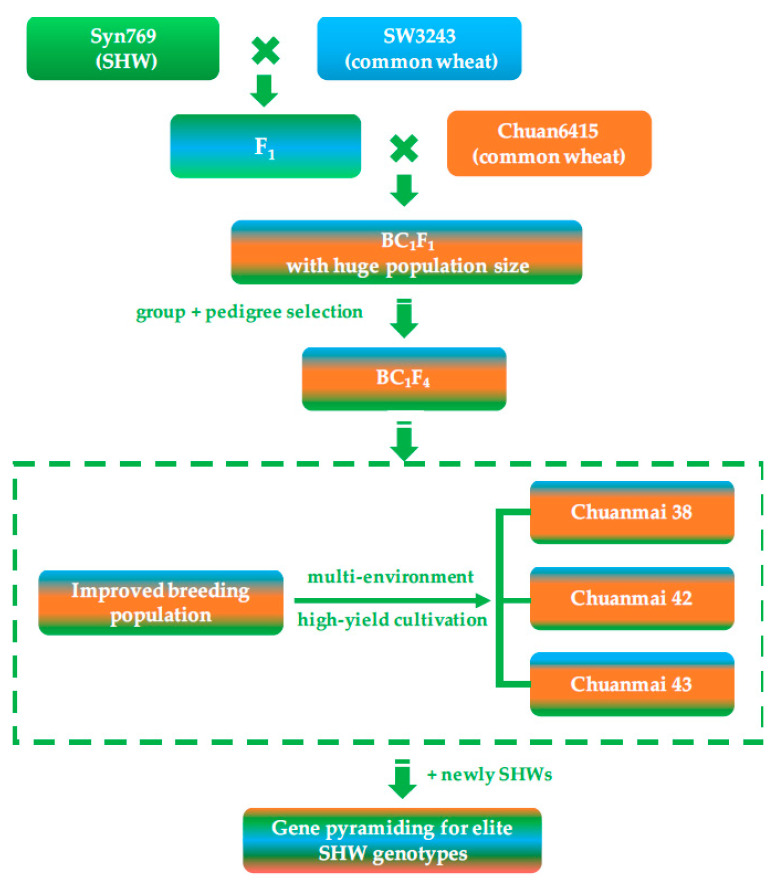
Direct application of SHW and breeding procedure of Chuanmai 42.

**Figure 3 genes-14-00283-f003:**
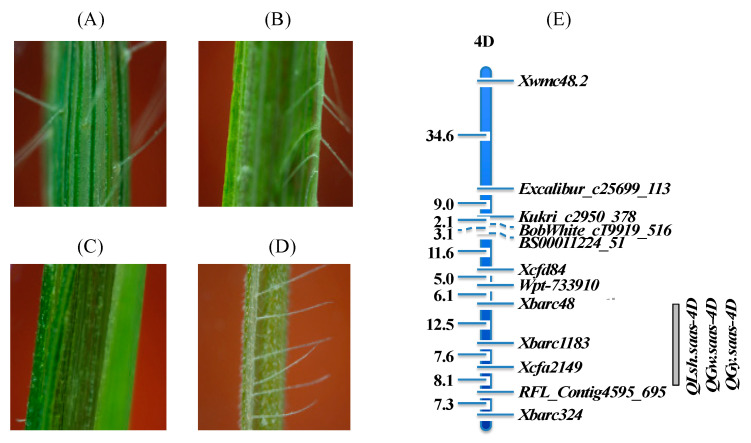
Hairy leaf sheaths from SHW and its association with grain yield. Leaf sheaths of (**A**) Chuanmai 42, (**B**) Syn769, (**C**) common wheat, and (**D**) *Ae. tauschii*; (**E**) QTL mapping for LSH, and grain yield (GY) and weight (GW).

**Figure 4 genes-14-00283-f004:**
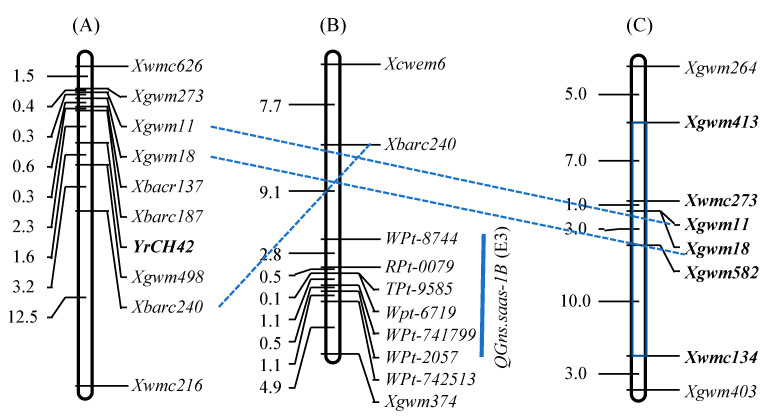
Molecular mapping of (**A**) *YrCH42*, (**B**) QTL for grain number per spike (GNS), and (**C**) the SHW alleles introgressed to Chuanmai 42 on chromosome 1B. Note: the interval in gray and the loci in bold italics were from Syn769.

**Figure 5 genes-14-00283-f005:**
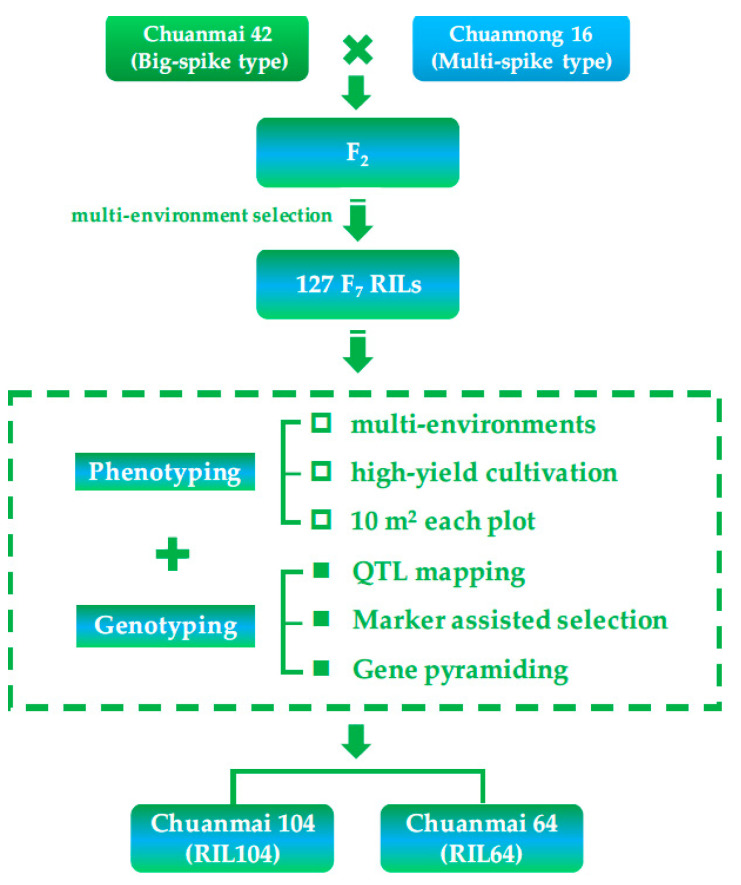
Application of SHW derivatives and breeding procedure of Chuanmai 104.

**Figure 6 genes-14-00283-f006:**
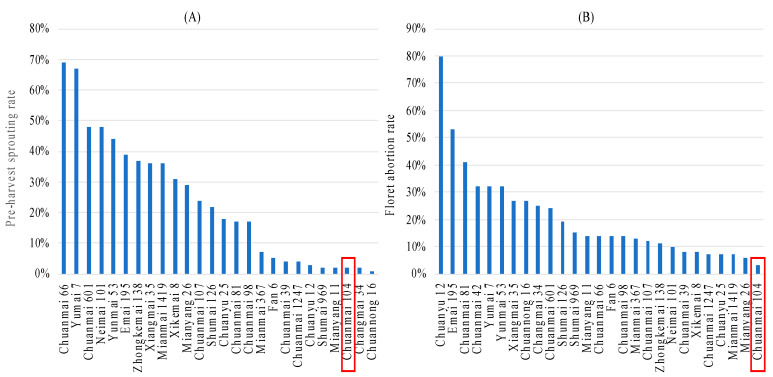
Pre-harvest sprouting (**A**) and low temperature (**B**) tolerance of Chuanmai 104.

**Figure 7 genes-14-00283-f007:**
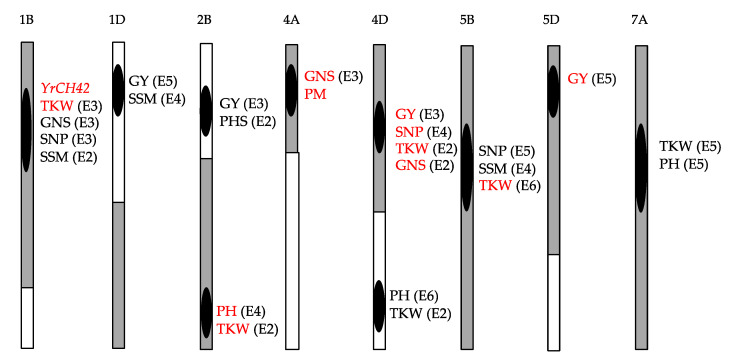
Genotypic patterns of Chuanmai 104 and yield-related QTLs on seven chromosomes. GY, grain yield; SSM, spike number per square meter; TKW, thousand-kernel weight; GNS, grain number per spike; SNP, spike number per plant; PH, plant height; PHS, pre-harvest sprouting resistance; PM, powdery mildew resistance. Red capital letters indicate that a QTL allele-increased phenotypic value was from the parent Chuanmai 42, and chromosome intervals in gray indicate the genomic regions from Chuanmai 42. The numbers in parentheses represent the number of environments in which each QTL could be detected.

**Table 1 genes-14-00283-t001:** Numbers of detected SNP/Indel between different generations of SHW and their parents ^¶^.

SHWs	Sub-Genome	Genomic Variation	S_0_	S_6_
Langdon/SQ783	AB genome	SNP	242,260	10,609,935
		Deletion	25,376	354,762
		Insertion	8017	346,855
		Total	275,653	11,311,552
	D genome	SNP	40,365	5,610,199
		Deletion	8121	263,781
		Insertion	8021	294,170
		Total	56,507	6,168,150
Langdon/SQ665	AB genome	SNP	39,185	235,260
		Deletion	6697	25,156
		Insertion	6674	7784
		Total	52,556	268,200
	D genome	SNP	27,691	3,906,319
		Deletion	4763	212,619
		Insertion	4764	186,499
		Total	37,218	4,305,437

^¶^ S_6_, the 6th selfing generation; S_0_ (F_1_), the first-generation hybrid.

**Table 2 genes-14-00283-t002:** Yield performance of Chuanmai 42 in Sichuan and national regional cultivar trials.

Year	Trial Grade	Average Yield (Kg/mu) ^a^	Check Cultivar	Increasing Rate (%)
2002	Sichuan regional cultivar trials	414.6	Chuanmai 28	70.2 ^b^
2003	Sichuan regional cultivar trials	403.2	Chuanmai 107	28.3
2003	National regional cultivar trials	354.7	Chuanmai 107	16.3
2004	National regional cultivar trials	406.3	Chuanmai 107	16.5

^a^ One mu is equal to 666.7 m^2^; ^b^ The low yield of this check cultivar in 2002 was due to its high susceptibility to stripe rust.

**Table 3 genes-14-00283-t003:** Cultivars released from the cross involving Chuanmai 42 in southwestern China.

Cultivar	Pedigree ^a^	Generation	Regional Trial Yield (Kg/mu)	Check Cultivar	Increasing Rate (%)	Released Time
Chuanmai 51	174/183//99-1572	1st	373.9	Chuanmai 107	13.4	2008
Chuanmai 56	Chuanmai 30/Chuanmai 42	1st	362.7	Chuanmai 107	13.5	2009
Chuanmai 58	Chuanmai 42/03Jian3//Chuanmai 42	1st	381.9	Mianmai 37	5.0	2010
Chuanmai 104	Chuanmai 42/Chuannong 16	1st	408.7	Chuanmai 42	8.5	2012
Shumai 969	SHW-L1/SW8188//Chuanyu 18/3/Chuanmai 42	1st	384.0	Mianmai 37	8.1	2013
Chuanmai 64	Chuanmai 42/Chuannong 16	1st	400.3	Mianmai 37	12.0	2013
Chuanmai 90	Jian38/99116//Chuanmai 42	1st	371.1	Mianmai 37	6.9	2014
Chuanmai 91	Neimai 8/Zhengmai 9023//00062/3/Chuanmai 42	1st	375.4	Mianmai 37	5.4	2014
Zhongkemai 138	Chuanmai 42/Chuanyu 16	1st	389.0	Mianmai 37	12.0	2014
Chuanmai 92	Neimai 8/Jian3//Chuanmai 42	1st	353.0	Mianmai 37	11.7	2015
Chuanmai 81	SW8019/99-1572//99-1572	1st	350.4	Mianmai 37	8.1	2015
Guohaomai 3	1227-185/99-1522//99-1572	1st	384.0	Chuanmai 42	−-3.5	2016
Chuanmai 602	Guinong 21/SW324/Chuanmai 42	1st	403.5	Mianmai 367	13.0	2017
Chuanmai 603	Aibaiguinong 21/Mianyang 26//Chuanmai 42	1st	355.1	Mianmai 367	6.3	2018
Chuanmai 604	Guinong 21/SW3243//Chuanmai 42	1st	388.2	Mianmai 367	13.0	2018
Chuanmai 86	R4117/1572	1st	382.8	Mianmai 367	12.5	2018
Neimai 101	99-1572/M0501	1st	394.8	Mianmai 367	10.5	2020
Chuanmai 53	477(99-1572)/Miannong 4//Y314	2nd	351.1	Chuanmai 107	20.2	2009
Chuanmai 66	99-1572/98-266//01-3570	2nd	377.3	Mianmai 37	5.6	2014
Chuanmai 67	99-1572/SW8688//01-3570	2nd	391.6	Mianmai 37	9.9	2014
Zhongkemai 169	Zhongkemai 138/Chuannong 27	2nd	380.7	Mianmai 367	7.8	2019
Zhongkemai 13	R64002/Zhongkemai 138	2nd	389.0	Mianmai 367	7.0	2020
Zhongke-NM 168	Zhongkemai 138/PW18	2nd	221.1	Yumai 13	2.3	2020
Chuanmai 68	99-1572/98-266//01-3570	2nd	379.6	Mianmai 37	16.5	2015
Chuanmai 601	Guinong 21/SW3243//Chuanmai 42/ Chuanmai 44	2nd	394.3	Chuanmai 42	5.8	2018
Xikemai 546	07Jian3401-05/Yumai 1	2nd	381.4	Mianmai 367	5.8	2021

^a^ 99-1572 (or 1572) were also Chuanmai 42.

**Table 4 genes-14-00283-t004:** Cultivars released from the cross involving Chuanmai 42 in southwestern China.

Cultivar	Pedigree ^a^	Generation	Regional Trial Yield (Kg/mu)	Check Cultivar	Increasing Rate (%)	Released Time
Chuanmai 69	Chuanmai 104/B2183	1st	361.6	Mianmai 37	13.2	2015
Chuanmai 1546	Chuanchongzu 104/Chuan 07005	1st	374.8	Mianmai 367	9.1	2018
Chuanmai 93	Pubing 3504/Chuanyu 20//Chuanmai 104	1st	385.7	Mianmai 367	16.9	2018
Chuanmai 96	Jian3/Chuannong 19//Chuanmai 104	1st	380.9	Mianmai 367	10.1	2018
Chumai 16	Neimai 8/Jian3//Chuanchongzu 104	1st	437.2	Yunmai 56	9.3	2018
Chuanmai 1580	Chuanchongzu 104/Chuan 07005	1st	394.3	Mianmai 367	10.4	2019
Chuanmai 98	Jian 3/Chuannong 19//Chuanmai 104	1st	405.9	Mianmai 367	13.0	2019
Chuanmai 1648	Chuanchongzu 104/CN16Xuan-1	1st	420.0	Mianmai 367	13.8	2020
Chuanmai 1603	Chuanchongzu 104/CN16Xuan-1	1st	417.6	Mianmai 367	15.9	2020
Chuanmai 1694	Chuanchongzu 104/Chuan 07005	1st	399.7	Mianmai 367	8.8	2020
Neimai 866	Chuanchongzu 104/Chuan 08Ping32	1st	390.3	Mianmai 367	7.5	2021

^a^ Chuanchongzu 104 were also Chuanmai 104.

## Data Availability

Not applicable.

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
