# Peer review of "Genetic Improvement and Application Practices of Synthetic Hexaploid Wheat"

_genes, 2023, doi:10.3390/genes14020283_

Round 1

Reviewer 1 Report

The manuscript by Wan et al reviewed the application practices of synthetic of synthetic hexaploid wheat in wheat genetic improvement and breeding, especially in Southwestern China. The reasons of why SHW has the potential to increase wheat yield were also discussed. The manuscript is of good quality, and I advise an acceptance subject to several minor revisions.

Title: Genetic improvement and application practices of synthetic hexaploid wheat in Southwestern China

Line 17 delete use, check others in the manuscript

Fig2 and 3, I don’t think these two figures are useful.

Line 131 its role as a major global crop

Line 151 Please mentioned here Chuanmai 42 was the first commercial SHW derivative in the world or in China.

Figure 5 delete or put it in the supplement

Line 180 check the spelling of LSH

Line 182 check the spelling of 2ed

Author Response

Point 1:Title: Genetic improvement and application practices of synthetic hexaploid wheat in Southwestern China

Response 1: On the surface, the cultivars derived from SHW was indeedly released in Southwestern China, and it is understandable to include the regional information. But no matter from what breeding platform the cultivars were released, the breeding strategy for application of SHW in our summary can be extended to any breeding platform of the world, and we insist that there is no need to add "in Southwestern China" in the title.

Point 2: Fig 2 and 3, I don’t think these two figures are useful

Response 2:We quite agree with this and we delete it without any hesitation (Line 112), and thanks for your advice.

Point 3: Please mentioned here Chuanmai 42 as the first commercial SHW derivative in the world or in China.

Response 3:We quite agree with this and we added this (Line 162).

Point 4: Figure 5 delete or put it in the supplement

Response 4:The content in the picture directly reflect the successful application practices of SHW as an officially technical evidence. And we think it is better to keep it.

other Points about spelling or handwriting have been checked, including SHW use, its role as a major global, crop spelling of LSH and 2nd.

Reviewer 2 Report

About Manuscript:  Genetic improvement and application practices of synthetic  hexaploid wheat (SHW)

This a review article aimed to present breeding strategy for application of SHW with the emphasis of wheat productivity in Chain. Overall, it seems that authors try to gather considerable information to address an issue in SHW. However there are some concerns as follow which should be consider.

1-Introduction suffer from citation of some new papers about SHWs such as:

Mokhtari, N., Majidi, M. M., & Mirlohi, A. (2022). Potentials of synthetic hexaploid wheats to improve drought tolerance. Scientific Reports, 12(1), 1-11.

Mourad, A. M., Morgounov, A., Baenziger, P. S., & Esmail, S. M. (2022). Genetic Variation in Common Bunt Resistance in Synthetic Hexaploid Wheat. Plants, 12(1), 2.

2-Improve language commanding of your work. Improve flow of thoughts almost in your contents of the work specially in discussion.

4- On of the most reason shifting to synthetic hexaploid wheat (SHW) is due to  representing wider and more comprehensive genetic base for abiotic stresses such a drought and salt. There is a weakness in this review about this issues in terms of phenotyping and genotyping.

5- Using new technologies specially NGS and GBS and GWAS in synthetic wheats can improve this review.

Author Response

Point 1: Introduction suffer from citation of some new papers about SHWs

Response 1: We think so, and added two new papers about SHWs: 64. Mokhtari, N.; Majidi, M.M.; Mirlohi, A. Potentials of synthetic hexaploid wheats to improve drought tolerance. Sci. Rep. 2022, 12, 20482 (Line 505); 65. Mourad, A.M.I.; Morgounov, A.; Baenziger, P.S.; Esmail, S.M. Genetic variation in common bunt resistance in synthetic hexa-ploid wheat. Plants 2023, 12, 2 (Line 507).

Point 2: Improve language commanding of your work. Improve flow of thoughts almostin your contents of the work specially in discussion.

Response 2: Actually, we have done our best to improve the language and writing.

Point 3: On of the most reason shifting to synthetic hexaploid wheat (SHW) is due to representing wider and more comprehensive genetic base for abiotic stresses such a drought and salt. There is a weakness in this review about this issues in terms of phenotyping and genotyping.

Response 3: We quite agree with you. And in our application practices of SHW for decades, we ignored the focus on the abiotic stresses resistance of SHW including drought and salt stress, mostly focused on higher grain yield and biotic stresses such as stripe rust, powdery mildew in terms of phenotyping and genotyping. And thanks for your advice.

Point 4:Using new technologies specially NGS and GBS and GWAS in synthetic wheats can improve this review.

Response 4: we added the NGS in Line 117

Reviewer 3 Report

The manuscript by Wan et al. describes wheat breeding be generating synthetic hexaploidy wheat. With significant examples of two elite cultivars with agronomical success in southwestern China, the manuscript efficiently illustrates the cultivar design, breeding histories and their further goal. The manuscript is generally well-written by scientific and formal English with a proper logical structure, and the figures are enough informative and appropriate. Therefore, it is suggested that the manuscript is enough qualified to be published on Genes. Followings are minor issues would be considered by authors for better quality.

Since the manuscript is focused in application of SHW in southwestern China, thus the title and/or abstract could include this regional information.

       L56; SPAD: Spelling-out an abbreviation when it appeared first would be helpful for the audience.

       Consider to refer a previous report (Gorafi et al. 2018 [https://doi.org/10.1007/s00122-018-3102-x]) of a parallel SHW approach.

Author Response

Point 1: Since the manuscript is focused in application of SHW in southwestern China, thus the title and/or abstract could include this regional information.

Response 1:On the surface, the cultivars derived from SHW was indeedly released in Southwestern China, and it is understandable to include the regional information. But no matter from what breeding platform the cultivars were released, the breeding strategy for application of SHW in our summary can be extended to any breeding platform of the world, and we insist that there is no need to add "in Southwestern China" in the title.

Point 2: SPAD: Spelling-out an abbreviation when it appeared first would be helpful for the audience.

Response 2: revised in Line 58: soil and plant analyzer development (SPAD) value

Point 3:Consider to refer a previous report (Gorafi et al. 2018) of a parallel SHW approach.

Response 3:The report by Gorafi et al. (2018) give us an effective strategy to evaluated the breeding value in more wider genetic donors of SHWs as multiple synthetic derivatives by genome‐wide association (GWA) analysis in a common wheat genetic background, which made utilization of SHW navigable. So we refer this report in Line 328.